# KAIROS: Scalable Model-Agnostic Data Valuation

Jiongli Zhu*, Parjanya Prajakta Prashant*, Alex Cloninger, Babak Salimi

University of California San Diego
jiz143@ucsd.edu, pprashant@ucsd.edu, acloninger@ucsd.edu, bsalimi@ucsd.edu

## Abstract

Data valuation techniques quantify each training example's contribution to model performance, providing a principled basis for data cleaning, acquisition, and selection. Existing valuation methods remain inadequate: *model-based* techniques depend on a single fitted model and inherit its biases, while *algorithm-based* approaches like Data Shapley scale poorly due to their need to train multiple models. Recent work has proposed model-agnostic alternatives based on Wasserstein distance between the training set and a clean reference set, but exact computation is expensive and approximations often misrank examples. We introduce KAIROS, a model-agnostic framework that values examples by their contribution to the Maximum Mean Discrepancy (MMD) between the training set and a clean reference distribution. Unlike Wasserstein methods, MMD admits a closed-form solution that requires no approximations and is scalable to large datasets. Additionally, KAIROS enables efficient online valuation: adding a new batch of $m$ examples requires only $O(mN)$ computation to update all scores, compared to $O(N^2)$ in prior work where $N$ is the training set size. Empirical evaluations on noise, mislabeling, and poisoning benchmarks show that KAIROS consistently outperforms state-of-the-art baselines in both accuracy and runtime. On ImageNet, KAIROS achieves up to $15 \times$ speedup over the fastest baseline while maintaining superior data valuation quality. Our results demonstrate that model-agnostic methods can match or exceed model-based approaches in performance while scaling to large datasets.

## 1 Introduction

The performance and behavior of AI systems significantly depend on the training data. The quality of data affects accuracy [63], robustness [41] and safety [9]. Data valuation methods, which quantify each training sample's contribution to model performance, are therefore essential for developing high-performance and reliable models. Existing valuation methods broadly fall into two categories. *Model-based* techniques, such as influence functions [33] and TracIn [47], evaluate the effect of individual data points on a single trained model, so their scores depend on the specific training run and hyperparameter settings [31]. *Algorithm-based* methods, e.g., Data Shapley [22], estimate each point's marginal contribution by averaging over many retrains, which becomes computationally infeasible for modern large-scale datasets [22, 27]. Neither category provides valuations that are both consistent and tractable on billion-example datasets.

Commercial and legal pressures amplify the need for model-agnostic data valuation methods. For large models, training datasets come from two primary sources: web-crawled data and purchased high-quality datasets. Web-crawled data is noisy, containing duplicates [38] and potential poisoning [25], making it essential to identify and filter harmful examples before training. High-quality licensed datasets, such as those from The Times [53] and Shutterstock [57], are expensive, requiring careful assessment of their value. Since training frontier models takes months and costs millions

---

* These authors contributed equally to this work.

39th Conference on Neural Information Processing Systems (NeurIPS 2025).

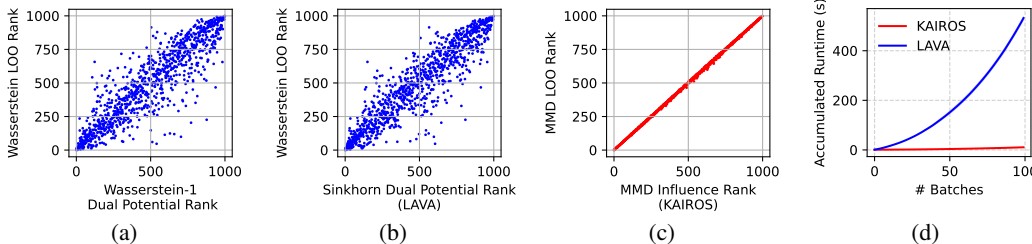

Figure 1: Comparison of Wasserstein- and MMD-based influence methods. (**a**) Leave-one-out (LOO) ranks versus dual potentials for unregularised optimal transport. (**b**) LOO ranks versus Sinkhorn dual potentials (LAVA). (**c**) LOO ranks versus MMD directional derivatives (KAIROS), which lie almost perfectly on the diagonal. Among the top-100 points, LAVA overlaps with true LOO rankings by 60% while KAIROS achieves 99% overlap. (**d**) Online runtime: KAIROS scales linearly with number of batches, whereas LAVA grows quadratically.

of dollars [11], companies typically train only once. This means data valuation must occur before training begins, ruling out model-based approaches. From the legal perspective, the EU AI Act Article 10 [1, 18] requires training data to be "relevant, sufficiently representative, and to the best extent possible, free of errors," with mandated assessment of data availability, quantity, and suitability. Critically, these requirements apply to the data itself, not to any specific model, making model-agnostic valuation essential for regulatory compliance. This need has motivated model-agnostic techniques such as LAVA [30] that value data before model training.

In this work, we introduce KAIROS, a scalable framework for model-agnostic data valuation. Each training example receives a *distributional influence score*: the change it induces in statistical divergence between the empirical distribution $P$ and a trusted reference $Q$ when that example is removed. To avoid prohibitive leave-one-out retraining costs, we define this score as the directional derivative of the divergence with respect to infinitesimal up-weighting. For popular divergences like Wasserstein-1, computing this derivative exactly is infeasible. Existing methods approximate it using Kantorovich dual potentials or its entropically-regularised Sinkhorn variant, as in LAVA [30]. We show these approximations produce rankings that drift from the true LOO ordering, over-valuing some points and under-valuing others (Figures 1a and 1b).

KAIROS instead uses Maximum Mean Discrepancy (MMD) as the divergence metric. We derive a closed-form directional derivative for MMD that matches LOO rankings (Figure 1c) and naturally incorporates label information. Our main contributions include:

1. We provide a closed-form influence function for MMD with $O(1/N^2)$ error from true leave-one-out valuations.

2. Computing influence scores requires no model training or iterative optimization, taking only $O(mN)$ time and $O(N)$ memory for batch size $m$. This enables efficient valuation for web-scale data, especially in the online setting (Figure 1d).

3. We establish that KAIROS satisfies symmetry and density-separation axioms, ensuring fair rankings for points that contribute equally and clear separation of low- and high-quality data.

4. The framework extends to conditional MMD kernels, allowing KAIROS to incorporate label information for supervised tasks.

5. Extensive experiments on label-noise, feature-noise, and back-door benchmarks demonstrate that KAIROS (i) more accurately detects corruptions, (ii) preserves accuracy when pruning low-value points and sharply degrades it when removing high-value points, and (iii) runs up to $50\times$ faster than prior methods LAVA and KNNSHAPLEY. Code is available at Github.

## 2 Related Work

Data valuation methods broadly fall into three categories: model-based, algorithm-based, and model-agnostic approaches. Model-based methods include influence functions [33, 24, 8], trajectory-based approaches [47, 4, 68], and post-hoc kernel approximations like TRAK [46] and DAVINZ [71]. These methods produce model-specific valuations that may vary across different models trained on the same data. Algorithm-based methods, particularly Data Shapley [22], define value in terms of marginal contributions to specific learning algorithms, with various approximations developed to avoid the

prohibitive cost of training $2^N$ models [28, 27, 54, 21, 68, 35, 67]. For bagging models, Data-OOB and 2D-OOB [36, 61] provide efficient alternatives using out-of-bag estimation. Algorithm-agnostic approaches like LAVA [30] quantify contributions based on dataset distance but suffer from poor performance on label error detection [29], computational complexity of $O(n^2)$, indeterministic approximations, and memory issues, which SAVA [32] partially addresses through batching strategies for Wasserstein computation. A detailed literature review is provided in Appendix A.

# 3 Data Valuation via Distributional Influence

We cast data valuation in the framework of distributional sensitivity analysis. As in standard settings, we are given a training set $D^{\text{train}} = \{(x_i, y_i)\}_{i=1}^{n_{\text{train}}}$ drawn i.i.d. from a noisy distribution $Q$, and a validation set $D^{\text{val}} = \{(x_i, y_i)\}_{i=1}^{n_{\text{val}}}$ from a clean target distribution $P$. For a candidate point $(x, y)$, we ask: *how much does it contribute to the distance between $P$ and $Q$?* Concretely, we quantify the value as the change in distributional distance when the empirical training distribution is infinitesimally perturbed toward the Dirac mass at $(x, y)$. Since this influence depends only on $P$, $Q$, and a distributional distance $d(P, Q)$, it defines an algorithm-agnostic valuation score.

The remainder of this section formalizes this idea (§3.1), derives a closed-form score using MMD (§3.2), extends it to labels using a conditional extension of MMD (§3.3), and presents an efficient algorithm for its computation (§3.4).

## 3.1 Distributional Distances

In the absence of a learning algorithm, the distributional distance between $P$ and $Q$ serves as a surrogate for the train–validation risk gap. Several distance measures such as Total Variation, Wasserstein distance [65, 66], and MMD [23] have been shown to upper bound the absolute difference between training and validation losses [5, 42, 51, 12]. We therefore value a training point by (the negative of) its contribution to the distance measure [30]. Specifically, we define this contribution by the influence function [37] of the distributional distance.

**Definition 1** (Distributional influence). *Let $d : \mathcal{M} \times \mathcal{M} \to \mathbb{R}$ be Gateaux-differentiable on the space $\mathcal{M}$ of probability measures. The influence of a point $x$ is*

$$\text{IF}_d(x; P, Q) = -\frac{d\big(P, (1-\varepsilon)Q + \varepsilon\delta_x\big) - d(P, Q)}{\varepsilon}\bigg|_{\varepsilon \to 0^+}, \tag{1}$$

*where $\delta_x$ denotes the Dirac measure at $x$.*

For finite samples, the influence function approximates the leave-one-out valuation with an error of $O(1/n_{\text{train}}^2)$ [64, Chapter 20, p. 291].

**Choosing a distance.** A distance $d$ suitable for data valuation must (i) admit a tractable influence formula and (ii) upper bound the train–validation loss gap. $f$-divergences form an important family of distance measures, which include popular metrics such as Kullback–Leibler (KL) divergence [14, 2]. However, they require density-ratio estimation (unstable in high dimensions [60]) and are not well-defined whenever the support of $P$ extends beyond that of $Q$. Integral Probability Metrics (IPMs) [44, 58] avoid these pitfalls by not requiring density ratios in their definition or computation. Given two distributions $P$ and $Q$, the IPM is defined as $d(P, Q) = \sup_{f \in \mathcal{F}} \big|\mathbb{E}_P[f] - \mathbb{E}_Q[f]\big|$, for a suitable function class $\mathcal{F}$, where $f$ is called the *critic function* that aims to distinguish between $P$ and $Q$. IPMs encompass various distance measures including Wasserstein-1 [65], MMD [23], and Total Variation Distance [44].

**IPM Influence decomposition.** Substituting the IPM into the influence definition (Eq. (1)) gives

$$\text{IF}_{\text{IPM}}(x; P, Q) = -\frac{\sup_{f \in \mathcal{F}}\big[\mathbb{E}_P[f] - \mathbb{E}_{(1-\varepsilon)Q + \varepsilon\delta_x}[f]\big] - \sup_{f \in \mathcal{F}}\big[\mathbb{E}_P[f] - \mathbb{E}_Q[f]\big]}{\varepsilon}\bigg|_{\varepsilon \to 0^+} \tag{2}$$

$$= \underbrace{\frac{\mathbb{E}_Q[f_\varepsilon^\star - f^\star] - \mathbb{E}_P[f_\varepsilon^\star - f^\star]}{\varepsilon}\bigg|_{\varepsilon \to 0^+}}_{\text{(i) critic-shift term}} + \underbrace{\big(f^\star(x) - \mathbb{E}_Q[f^\star]\big)}_{\text{(ii) point-gap term}}. \tag{3}$$

Let $f^\star$ and $f_\varepsilon^\star$ denote the optimal critics before and after up-weighting a point $x$ by an infinitesimal mass $\varepsilon$. The *critic-shift* term in the IPM influence function depends on $x$ and is generally intractable. If the difference $(f_\varepsilon^\star - f^\star)$ decays as $O(\varepsilon^\alpha)$ for some $\alpha > 1$, the critic-shift term vanishes and Equation (3) collapses to the simpler *point-gap* term. However, this decay need not hold for all IPMs; in particular, it does not hold for Wasserstein-1 [39, p. 39-40]. In the Wasserstein-1 case, $f^*$

corresponds to a Kantorovich potential, which is *non-unique* [66, 59, 16]. Discarding the critic-shift term therefore yields influence values that are non-deterministic.

Current model-agnostic data-valuation methods like LAVA [30] rely on this simplification, retaining only the point-gap term for the Wasserstein-1 metric. This introduces two significant issues. (i) Since the dual critic is non-unique, the resulting influence values can vary arbitrarily, violating determinism. (ii) To enhance scalability, LAVA replaces the exact Wasserstein-1 objective with its Sinkhorn-regularized counterpart; however, this introduces a bias of order $O\big(d\,\nu\log(1/\nu)\big)$, where $d$ is the data dimensionality and $\nu$ is the regularization strength [20]. Correcting this bias is computationally expensive, and even as $\nu \to 0$, the neglected critic-shift term remains unresolved, causing the resulting influence scores to deviate substantially from the true leave-one-out rankings, the stated objective of the method (Figure 1b).

## 3.2 Closed-form Influence via MMD

We address these challenges by using MMD [23], an IPM with a closed-form expression for the distance. *We derive a closed-form influence function that can be computed directly without first computing the distance itself.* This section focuses on the marginal distribution of features. We incorporate label information in subsection 3.3. Given two distributions $P$ and $Q$, the MMD distance between them is defined as:

$$\text{MMD}(P,Q) = \sup_{\|f\|_{\mathcal{H}} \leq 1} \big(\mathbb{E}_{x \sim P}[f(x)] - \mathbb{E}_{x \sim Q}[f(x)]\big) = \|\mu_P - \mu_Q\|_{\mathcal{H}}, \tag{4}$$

where $\mu_P = \mathbb{E}_{x \sim P}[\phi(x)]$ and $\mu_Q = \mathbb{E}_{x \sim Q}[\phi(x)]$ are the kernel mean embeddings in the RKHS $\mathcal{H}$, and the kernel $k(x, x') = \langle \phi(x), \phi(x') \rangle_{\mathcal{H}}$ [23]. While computing $\|\mu_P - \mu_Q\|_{\mathcal{H}}$ directly can be challenging, the squared distance admits a closed-form expression via the kernel trick:

$$\text{MMD}^2(P,Q) = \mathbb{E}_{x,x' \sim P}[k(x,x')] + \mathbb{E}_{x,x' \sim Q}[k(x,x')] - 2\mathbb{E}_{x \sim P, x' \sim Q}[k(x,x')]. \tag{5}$$

This enables a closed-form computation of the influence with respect to MMD via the chain rule.

**Proposition 1.** *The influence function for* MMD *as the distance metric is, up to additive and positive multiplicative constants, given by*

$$\text{IF}_{\text{MMD}}(x; P, Q) = \mathbb{E}_{x' \sim P}[k(x', x)] - \mathbb{E}_{x' \sim Q}[k(x', x)]. \tag{6}$$

The full derivation is provided in Appendix B.1. Note that for downstream tasks such as feature error detection, backdoor attack identification, or ranking the most and least valuable points, only the relative rankings are important [35, 67]; hence, the additive and multiplicative constants in the influence function do not affect the outcome. Henceforth, we use $\text{IF}(\cdot)$ to denote the rescaled version of $\text{IF}_{\text{MMD}}(\cdot; P, Q)$, omitting $P$ and $Q$ for brevity. For a training point $x_i \in D^{\text{train}}$, the unbiased finite-sample estimate of its MMD-based influence is given by:

$$\widehat{\text{IF}}(x_i) = \frac{1}{n_{\text{val}}} \sum_{j=1}^{n_{\text{val}}} k(x_j^{\text{val}}, x_i) - \frac{1}{n_{\text{train}} - 1} \sum_{j=1, j \neq i}^{n_{\text{train}}} k(x_j^{\text{train}}, x_i), \tag{7}$$

where the first term approximates $\mathbb{E}_{x' \sim P}[k(x', x_i)]$ using validation data and the second term approximates $\mathbb{E}_{x' \sim Q}[k(x', x_i)]$ using the training data excluding $x_i$.

**Properties.** Beyond closely approximating leave-one-out scores [64], our MMD-based influence enjoys two nice properties: *(i) Symmetry* ensures that points making the same marginal contribution to the $\widehat{\text{MMD}}$ (the MMD estimate) receive identical influence, yielding fair rankings; *(ii) Density-separation* guarantees the existence of a global threshold that cleanly partitions regions where the validation distribution dominates ($P > Q$) from those where the training distribution dominates ($Q > P$), enabling high-accuracy in detecting noisy samples and backdoor attacks.

**Proposition 2** (Symmetry). *Let $D^{\text{train}}$ and $D^{\text{val}}$ be finite samples from distributions $Q$ and $P$, respectively. If for all subsets $S \subseteq D^{\text{train}} \setminus \{x_i, x_j\}$,*

$$\widehat{\text{MMD}}(D^{\text{val}}, S \cup \{x_i\}) - \widehat{\text{MMD}}(D^{\text{val}}, S) = \widehat{\text{MMD}}(D^{\text{val}}, S \cup \{x_j\}) - \widehat{\text{MMD}}(D^{\text{val}}, S)$$

*then $\widehat{\text{IF}}(x_i) = \widehat{\text{IF}}(x_j)$.*

See Appendix B.2 for proof.

**Proposition 3** (Density Separation). *Let $P$ and $Q$ be two probability distributions on $\mathcal{X} \subseteq \mathbb{R}^n$. For any $\epsilon > 0$ and $r > 0$, there exists a Gaussian isotropic kernel $k$ such that for $\mathrm{IF}(x) = \mathbb{E}_{x' \sim P}\big[k(x, x')\big] - \mathbb{E}_{x' \sim Q}\big[k(x, x')\big]$, the following holds*

$$\begin{cases} \mathrm{IF}(x) > 0, & \text{if } P(x') - Q(x') \geq \epsilon \,\forall\, x' \in B(x, r), \\ \mathrm{IF}(x) < 0, & \text{if } P(x') - Q(x') \leq -\epsilon \,\forall\, x' \in B(x, r), \end{cases}$$

*where $B(x, r) = \{x' : \|x' - x\| < r\}$.*

In Appendix B.3 we give a full proof of Proposition 3, and we also present a real-world example illustrating its practical effect, where the MMD-based influence scores for clean versus corrupted points are perfectly split by a near-zero threshold, whereas the Wasserstein-based scores exhibit substantial overlap and admit no such clean cutoff.

### 3.3 Capturing Feature–Label Correlations with MCMD

Marginal MMD over $P(X)$ and $Q(X)$ effectively detects covariate shift but overlooks label-specific corruptions—such as flipped labels, back-door triggers, or concept drift—that alter $P(Y\,|\,X)$ without changing $P(X)$. Directly operating on the joint distribution $(X, Y)$ using a product kernel often reduces sensitivity to label noise: in high-dimensional feature spaces, the geometry of $X$ dominates the kernel distances, causing small perturbations in labels $Y$ to barely shift the joint embedding, and diminishing the test's power as dimensionality grows [49, 50]. We therefore retain marginal MMD for detecting feature-level anomalies, and augment it with the expected value of Maximum Conditional Mean Discrepancy (MCMD) [52, 45], a conditional extension of MMD denoted as E-MCMD. This hybrid criterion enables KAIROS to identify both covariate and label anomalies within a unified, kernel-based theoretical framework.

**Definition 2** (Maximum Conditional Mean Discrepancy (MCMD) [52, 45]). *The MCMD between conditional distributions $P(Y|X)$ and $Q(Y|X)$ at point $x$ is:*

$$\mathrm{MCMD}_{P,Q}(x) := \|\mu_{Y|X}^P(x) - \mu_{Y|X}^Q(x)\|_{\mathcal{H}_{\mathcal{Y}}},$$

*where $\mu_{Y|X}^P(x) = \int \phi(y)\, dP(y|x)$ and $\mu_{Y|X}^Q(x) = \int \phi(y)\, dQ(y|x)$ are known as the conditional kernel mean embeddings.*

To aggregate this measure across covariate space, we take the expectation of $\mathrm{MCMD}(\cdot)$ over the training distribution i.e. $\mathrm{E\text{-}MCMD}(P, Q) = \mathbb{E}_{x \sim Q}[\mathrm{MCMD}_{P,Q}(x)]$. We derive the influence for E-MCMD via Definition 1.

**Proposition 4.** *The influence function for E-MCMD as the distance metric is, up to additive and positive multiplicative constants, given by*

$$\mathrm{IF}_{\text{E-MCMD}}(x, y; P, Q) = -\|\mu_{Y|X}^P(x) - \phi(y)\|_{\mathcal{H}_{\mathcal{Y}}} \tag{8}$$

The full derivation is provided in Appendix B.4. Henceforth, we use $\mathrm{IF}_{cond}(x, y)$ to denote the rescaled version of $\mathrm{IF}_{\text{E-MCMD}}((x, y); P, Q)$, omitting $P$ and $Q$ for brevity. We can simplify this expression when $Y$ is categorical ($Y \in \{0, 1, \ldots, C - 1\}$). For categorical labels, consider the feature map $\phi(y) = e_y \in \mathbb{R}^C$ and the kernel $k(y, y') = \mathbb{1}\{y = y'\}$ where $e_y$ is the $y$-th standard basis vector. For this kernel and mapping function, we have $\mu_{Y|X}^P(x) = [P(0|x), \ldots, P(C - 1|x)]^T$ which is simply the probability vector for each class conditioned on $x$. Therefore:

$$\mathrm{IF}_{\text{cond}}(x, y) = -\|\mu_{Y|X}^P(x) - e_y\|_2 = -\sqrt{\sum_{y' \neq y} P(y'|x)^2 + (P(y|x) - 1)^2}. \tag{9}$$

For the finite sample case, $P(Y|X)$ can be estimated by a classifier trained on $D^{\text{val}}$ that returns $\hat{P}(y|x)$. The finite sample estimator for the conditional influence is:

$$\widehat{\mathrm{IF}}_{\text{cond}}(x, y) = -\sqrt{\sum_{y' \neq y} \hat{P}(y'|x)^2 + (\hat{P}(y|x) - 1)^2}. \tag{10}$$

**Combined Net Distance and Influence.** Finally, we integrate both marginal and conditional discrepancies into a single "net" distance:

$$d_{\text{net}}(P, Q) = (1 - \lambda)\, \mathrm{MMD}(P_X, Q_X) + \lambda\, \mathrm{E\text{-}MCMD}(P, Q), \tag{11}$$

where $\lambda > 0$ balances the two terms. The overall influence of $(x, y)$ on $d_{\text{net}}$ is simply

$$\mathrm{IF}_{\text{net}}(x, y) = (1 - \lambda)\, \mathrm{IF}(x) + \lambda\, \mathrm{IF}_{\text{cond}}(x, y). \tag{12}$$

**Generalization Error Bound.** We establish a theoretical link between the net discrepancy $d_{\text{net}}$ and downstream model performance. Under mild regularity assumptions, the expected train–validation loss gap is bounded above by the sum of the marginal MMD and conditional E-MCMD. Consequently, removing a point which decreases this distance provably tightens the out-of-distribution error bound for any learning algorithm. The influence is a first-order approximation of the effect of removing a point, suggesting that removing points with large influence scores could decrease the out-of-distribution error. A concise statement follows, with the complete theorem and proof in Appendix B.5.

**Theorem 1** (Bounding transfer loss (simplified)). *Let $(\mathcal{X}, d_{\mathcal{X}})$ and $(\mathcal{Y}, d_{\mathcal{Y}})$ be compact metric spaces and $\mathcal{Z} = \mathcal{X} \times \mathcal{Y}$. Let $L : \mathcal{Z} \to \mathbb{R}$ be a continuous loss function. Then, for some constant c,*

$$\mathbb{E}_{(x,y) \sim Q}[L(x,y)] \leq \mathbb{E}_{(x,y) \sim P}[L(x,y)] + c\Big( \text{MMD}(P_X, Q_X) + \mathbb{E}_{x \sim Q_X}\big[\text{MCMD}_{P,Q}(x)\big]\Big).$$

### 3.4 Batch Computation and Streaming Updates

Modern ML pipelines often require continuous data valuation as new data arrives. Examples include language models trained on fresh web crawls or recommender systems processing new user interactions. Recomputing influence scores from scratch after each update is prohibitively expensive, scaling as $O(N^2)$ with dataset size. We address this with a two-phase approach. The *offline initialization* scans the initial training and validation sets once, computing and caching three quantities per training point: the average training kernel similarity $\mathcal{A}_i$, the average validation kernel similarity $\mathcal{B}_i$, and the label residual $\mathcal{R}_i$. The *online update* then processes each incoming batch efficiently. For new points, we compute their statistics from scratch. For existing points, we update only $\mathcal{A}_i$ using kernel evaluations between old and new points. This approach maintains accurate valuations in streaming settings without quadratic recomputation.

**Offline Algorithm.** In the standard offline setting, we assume access to the entire dataset and a classifier trained on $D^{\text{val}}$ that provides predicted probability vectors $\hat{y}_i^{\text{train}} = \hat{P}(y|x_i)$ for training points. For each training point $(x_i, y_i) \in D^{\text{train}}$, we precompute three key quantities:

$$\mathcal{A}_i = \frac{1}{n_{\text{train}} - 1} \sum_{\substack{j=1 \\ j \neq i}}^{n_{\text{train}}} k(x_j^{\text{train}}, x_i), \quad \mathcal{B}_i = \frac{1}{n_{\text{val}}} \sum_{j=1}^{n_{\text{val}}} k(x_j^{\text{val}}, x_i), \quad \mathcal{R}_i = \|y_i^{\text{train}} - \hat{y}_i^{\text{train}}\|_2 \tag{13}$$

Using these precomputed values, the feature influence is $\widehat{\text{IF}}(x_i) = \mathcal{B}_i - \mathcal{A}_i$ (from Equation (7)) and the conditional influence is $\widehat{\text{IF}}_{\text{cond}}(x_i, y_i) = R_i$ (from Equation (10)). Therefore, the final net influence score is:

$$\widehat{\text{IF}}_{\text{net}}(x_i, y_i) = (1 - \lambda)(\mathcal{B}_i - \mathcal{A}_i) + \lambda \mathcal{R}_i \tag{14}$$

The time complexity for this algorithm is $O(n_{\text{train}}^2 + n_{\text{train}} \cdot n_{\text{val}})$, which simplifies to $O(n_{\text{train}}^2)$ since typically $n_{\text{train}} > n_{\text{val}}$.

**Online Algorithm.** In the online setting, we process data in batches. At time $t$, we receive a new batch of size $m$, bringing the total dataset size to $n_{t+1} = n_t + m$. We update the influence scores for both existing points and the new batch. While a naive approach of recomputing everything from scratch would require $O(n_{t+1}^2)$ time, our elegant influence expression allows computation in $O(n_t \cdot m + m^2)$.

For existing points $(x_i, y_i)$ where $i \leq n_t$, the terms $\mathcal{B}_i$ and $\mathcal{R}_i$ remain unchanged. The training kernel average $\mathcal{A}_i$ can be efficiently updated as:

$$\mathcal{A}_i^{(t+1)} = \frac{1}{n_{t+1} - 1} \left( (n_t - 1) \cdot \mathcal{A}_i^{(t)} + \sum_{j=1}^{m} k(x_{n_t+j}^{\text{train}}, x_i) \right)$$

Note that this update requires computing only $m$ new kernel evaluations per existing point, not $n_{t+1}$. For new points $(x_{n_t+i}, y_{n_t+i})$ where $i = 1, \ldots, m$, we compute all three quantities from scratch using Equation (13). The final influence scores are then computed as $\widehat{\text{IF}}_{\text{net}}(x_i, y_i) = \lambda(\mathcal{B}_i - \mathcal{A}_i) + (1 - \lambda)\mathcal{R}_i$ for all points. This update procedure runs in $O(n_t \cdot m)$ for the old data and $O(m^2)$ for the new batch resulting in total complexity of $O(n_t \cdot m + m^2)$. Detailed algorithm provided in Appendix C.

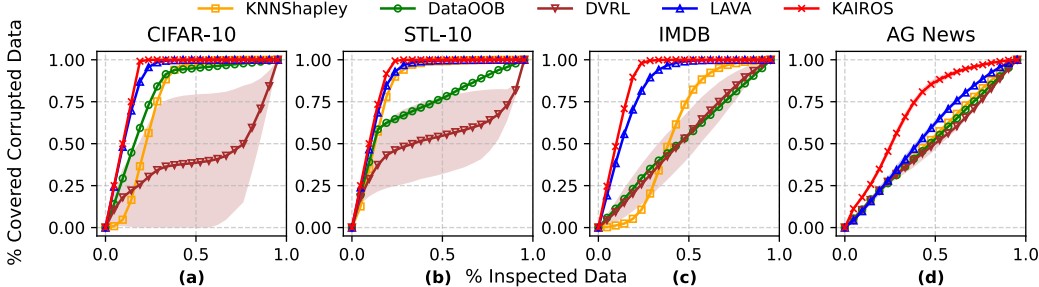

Figure 2: Feature noise detection results.

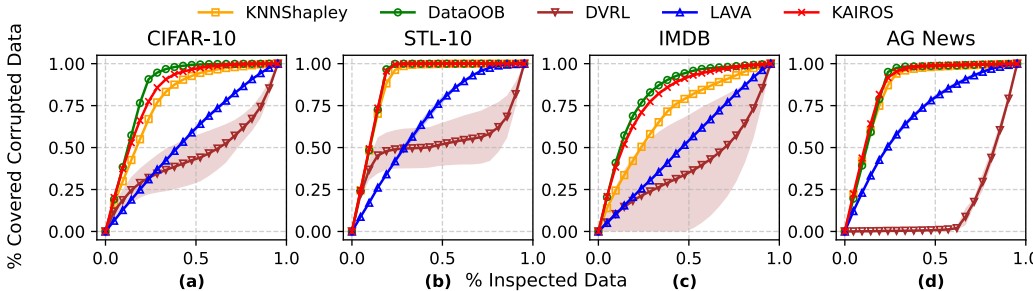

Figure 3: Label noise detection results.

## 4 Experiments

We implement KAIROS as a custom data valuation method in the OpenDataVal benchmark [29], a benchmark for evaluating data valuation methods. We evaluate all methods on three key applications: (1) detecting noisy data, mislabels, and mixture of them, (2) detecting malicious data injected by data poisoning attacks, and (3) data pruning by removing data with the lowest values. We also test removing data with the highest values to thoroughly examine the effectiveness of all data values, including low and high. In addition, we test the runtime of methods with varying data sizes, in offline and online settings. Experiments are repeated five times with different random seeds, and we report the mean (colored regions denote the standard deviations). We include additional experiments on the hyperparameter and kernel choice, noisy validation set, and million-scale data in Section D.

**Datasets.** We evaluate on four widely used datasets, including CIFAR-10 [34], STL-10 [10], IMDB [43], and AG News [77], to cover both image and text modalities. In most experiments, we simulate limited clean-data availability by using 10000 noisy training examples and 300 clean validation examples, with a held-out test set of 10000 clean samples. For the smaller STL-10 dataset, we scale down to 3700 training, 300 validation, and 1000 test examples.

**Baselines and Hyper-parameters.** We compare KAIROS with four state-of-the-art data valuation methods with different mechanisms: LAVA [30], DATAOOB [36], DVRL [76], and KNNSHAP-LEY [27]. For KAIROS, we set the Gaussian kernel bandwidth to the median of all pairwise distances and fix the balancing factor in Equation (12) to 0.03. See details in Appendix D.

**Noise and Mislabel Detection.** In this experiment, we introduce noise into 20% of the data. Following [29, 30, 73, 72], we inject feature noises by adding white noise to the images and randomly replacing words with other words for texts, and introduce label noises by randomly changing the labels of corrupted samples to other classes. Figures 2 and 3 present the performance of different data valuation methods in identifying corrupted samples stemming from two distinct sources of noise: feature perturbations and mislabels. Each curve plots the cumulative fraction of corrupted data recovered as a function of the percentage of training data inspected.

KAIROS consistently achieves strong performance across both noise types and all datasets. In the feature noise setting (Figure 2), it ranks noisy samples more effectively than all baselines on all datasets, especially in the early inspection phase. In particular, on AG News, all methods except KAIROS are close to the diagonal, meaning that they perform similarly to assigning random values

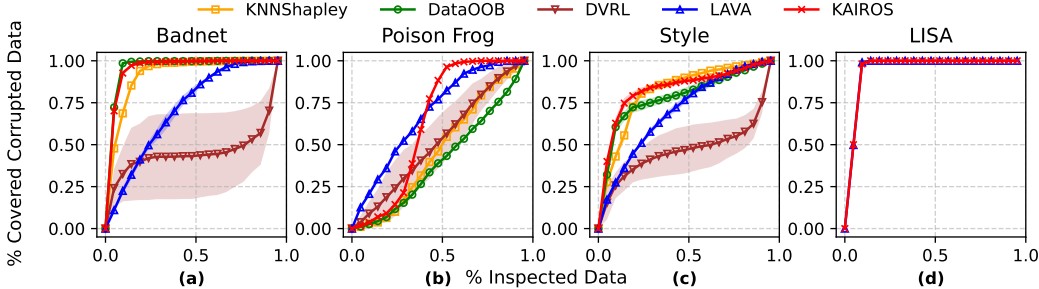

Figure 4: Malicious data detection results.

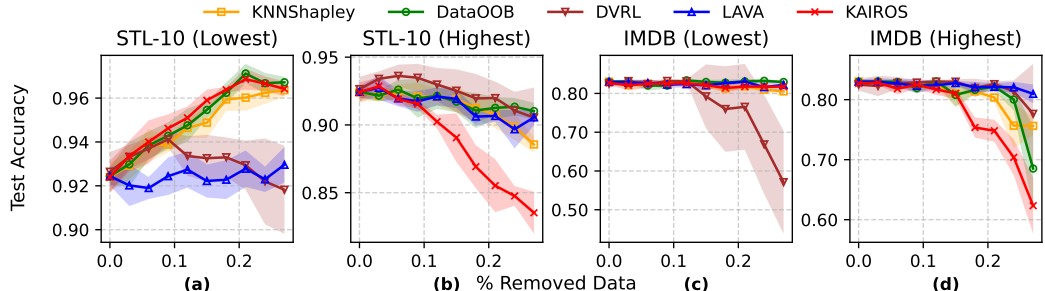

Figure 5: Effect of removing the least valuable (a,c) and the most valuable (b,d) data points on test accuracy.

to all data points. In the presence of label noises (Figure 3), our method remains competitive and outperforms most baselines, particularly on AG News and STL-10.

Overall, while some methods show competitive performance on specific datasets or noise types, such as LAVA on feature noises and DATAOOB on label noises, they fail to generalize well across both feature and label noise scenarios. In contrast, KAIROS archives top-1 detection accuracy in 6 (out of 8) scenarios, and stays top-2 for the remaining 2 cases, indicating its versatile performance in noise detection.

**Malicious Data Detection.** We evaluate robustness under adversarially crafted poisoning attacks following [30]. We test Badnet [25] and poison frogs [55] attacks on CIFAR-10, and clean label style attack [48] and LISA [26] on AG News. For CIFAR-10 attacks, we inject 3% malicious data; for AG News attacks, we inject 10% poisonous data. Model-based techniques are not applicable for LISA fine-tuning of LLMs due to the computational expense of training hundreds of models.

Across the four scenarios in Figure 4, LAVA underperforms on scenarios (a) and (c), while DATAOOB struggles with (b,c) and cannot handle (d). KNNSHAPLEY, DATAOOB, and DVRL show less competitive performance than KAIROS across all cases. LAVA achieves comparable performance with KAIROS only in detecting poisoned fine-tuning data but is significantly less effective for Badnet and style attacks.

**Point Removal.** Figure 5 (a–d) shows how test accuracy changes when removing the least or most valuable training data identified by each method. We use 20% mislabeled data. Good valuations should cause large accuracy drops when removing high-value data and small drops (or increases) when removing low-value data.

In both scenarios, our method yields the most desirable behavior. When pruning the least-valuable data (Figure 5 (a, c)), on STL-10, KAIROS, DATAOOB, and KNNSHAPLEY increase the test accuracy by a similar amount, while the values obtained from DVRL and LAVA do not help. On IMDB data, all methods except DVRL keep the test accuracy when removing 30% of the data. For both datasets, when discarding the most valuable data (Figure 5 (b, d)), KAIROS results in more significant accuracy drops than all baselines. This indicates that KAIROS gives both meaningful low and high values to the data, while most baselines only effectively identify low-valued data.

**Effect of Validation Sample Size.** In practice, validation sets are often small due to expensive labeling. To understand how many validation samples are needed to obtain reliable data values, we conduct noise detection with varying validation set sizes. In particular, to better compare the convergence of our method (based on MMD) and LAVA (based on Wasserstein), we test with 20% feature noises on IMDB data, the scenario where LAVA performs relatively well. To measure effectiveness, we adopt the detection accuracy, defined by the percentage of correctly

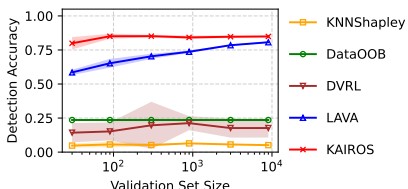

Figure 6: Label noise detection accuracy under varying validation set sizes.

identified corrupted data among the 20% least-valued data points. As shown in Figure 6, the variance of KAIROS, indicated by color regions, shrinks more quickly with growing validation set size than that of LAVA. Although LAVA benefits from an increased validation set, it requires 9K validation samples to reach an accuracy of 0.77. In contrast, KAIROS only requires 30 samples to achieve this. This implies that MMD is more robust and reliable compared to Wasserstein under finite samples.

**Offline Runtime.** To understand the scalability of different methods, we measure the runtime of methods on CIFAR-10 data with label noise. We vary the training set size and keep the static validation set size of 300. As shown in Figure 7, KAIROS, LAVA, and KNNSHAPLEY are similarly efficient, and are significantly faster than DVRL (10x) and DATAOOB (100x). Although DVRL's runtime grows slowly with # training samples, as shown in previous results, it performs poorly in most applications,

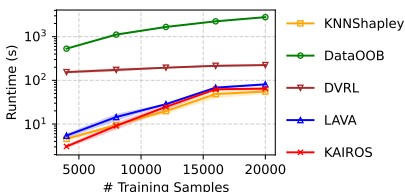

Figure 7: Offline runtime comparison.

making it not practically applicable. DATAOOB's high runtime results from training models on various bootstraps.

**Online Runtime.** To show the adaptability of KAIROS in the online setting, we split 10000 of CIFAR-10 data into 100 batches, each containing 100 samples, and feed them in a streaming way. We measure the accumulated time taken to conduct data valuation after each batch update. KAIROS adopts Algorithm 1 for value computation and updates, while LAVA and KNNSHAPLEY have no direct adaptation to this setting, thus have to re-calculate the values when a new batch comes in. DATAOOB and DVRL

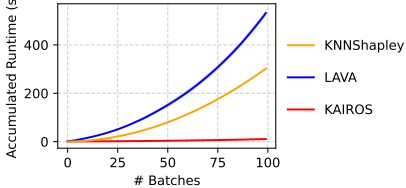

Figure 8: Online runtime comparison.

are omitted for this experiment as they take more than 8 hours to complete the experiment, meaning that they are not practical to use in this setting. As shown in Figure 8, KAIROS is significantly faster than LAVA and KNNSHAPLEY in the online setting. The speedup becomes more significant when more batches are included. The speedup reaches 28x compared to KNNSHAPLEY, and 50x compared to LAVA when all batches are fed.

**Large Scale Experiment on ImageNet.** To evaluate KAIROS at scale, we compare it to the scalable variant of LAVA, SAVA, on ImageNet [17], which contains 1.28M training images and 1000 classes. We extract features with a ResNet-50 encoder and run all methods on a single NVIDIA A100 (40GB VRAM). We report the runtime and the AUC of the "% inspected data vs. % corrupted data covered" curve (also shown in Figures 2 and 3) under both feature-noise and label-noise settings.

| Method | Runtime | AUC |
|--------|---------|-----|
| KAIROS | 7m56s | 0.869 |
| SAVA | 1h58m | 0.817 |

Table 1: Feature noise detection on ImageNet.

| Method | Runtime | AUC |
|--------|---------|-----|
| KAIROS | 7m52s | 0.861 |
| SAVA | 1h58m | 0.484 |

Table 2: Label noise detection on ImageNet.

Our results show that KAIROS significantly outperforms SAVA in both efficiency and effectiveness. The efficiency gain of KAIROS comes from its closed-form solution, which enables batch-based GPU acceleration and avoids expensive Sinkhorn computations. By contrast, SAVA (and LAVA) require computing pairwise conditional Wasserstein distances between $P(x \mid y)$ across all class pairs, leading to $\frac{1000 \times 999}{2}$ computations. This accounts for over 80% of their runtime, making KAIROS more suitable for large-scale applications.

**Summary.** Across all tasks and datasets, KAIROS consistently achieves performance gains over state-of-the-art baselines. It effectively ranks data under both natural and adversarial data corruptions and noises. The runtime experiments demonstrate the advantageous efficiency of KAIROS compared to baselines, especially in the practical online and large-scale settings.

## 5 Conclusions, Limitations and Broader Impacts

We introduce KAIROS, a scalable data valuation framework that uses Maximum Mean Discrepancy to compute closed-form influence functions for detecting feature noise, label corruption, and backdoors. KAIROS achieves up to $50\times$ speedup over existing methods with $O(mN)$ complexity in online settings, making it practical for web-scale deployment while maintaining faithful leave-one-out rankings. Our approach provides theoretical guarantees through symmetry and density separation properties and offers model-agnostic influence scores that enable transparent data quality assessment, fairness auditing, and regulatory compliance without requiring model retraining. Current limitations include the use of fixed kernels and a fixed balancing coefficient for all tasks. Future work should focus on learned kernels, efficient approximate methods, and regression extensions.

## 6 Acknowledgment

This research was supported by NSF award IIS-2340124, NSF CISE CIF award 2403452, and NIH grant U54HG012510. The views, opinions, and findings presented are those of the authors and do not necessarily represent those of the NSF or NIH.

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

# A  Related Work

**Model-based Data Valuation**   A common approach for data valuation is the leave-one-out (LOO) score, i.e., the change in performance when a point is removed from training, but this is computationally expensive. Influence functions [33] approximate these scores using second-order derivatives, though they remain intractable for modern deep models. Recent advances have made influence functions scalable to large models [24, 8], and others trace test loss over the training trajectory to attribute influence [47, 4, 68]. TRAK [46] takes a post-hoc approach, approximating the model as a kernel machine to trace predictions back to training examples, while DAVINZ [71] uses neural tangent kernel approximations to predict influence at initialization. Other approaches include reinforcement learning to learn data values [76], prediction from noisy labels [13], and dynamic self-weighting mechanisms within loss functions [70].

**Algorithm-based Data Valuation**   Unlike model-based methods that track influence during or after training, algorithm-based approaches define data value in terms of a specific learning algorithm and utility function. A major line relies on Shapley values [56], which uniquely satisfy fairness axioms such as symmetry and null-value. Data Shapley [22] defines a point's value as its average marginal contribution to model utility across all subsets, but exact computation requires training $2^N$ models. Efficient approximations [28], closed-form solutions for $k$-NN [27], class-wise extensions [54], and distributional variants [21] have been proposed. Retraining multiple models can be avoided through gradient- and hessian-based approximations [68]. However, the reliability of shapley methods is sensitive to the utility function [69]. Other work relaxes the efficiency axiom to improve robustness [35, 67]. For bagging models, Kwon et al. [36] show that out-of-bag estimates yield effective approximations.

**Algorithm-agnostic Data Valuation**   Algorithm-agnostic valuation methods operate without knowledge of the learning algorithm. Just et al. [30] propose LAVA, which quantifies the contribution of a point based on its contribution to the dataset distance. However, their method has several limitations: (i) it performs poorly in identifying label errors [29]; (ii) their approximations are not always aligned with the leave one out valuation and can be indeterministic [66] (see Figure 1b); and (iii) the method is computationally expensive (even in approximate form, has $O(n^2)$ complexity) [15]. The distance must be computed over the entire dataset even when valuing a single point (iv)the memory complexity is $O(n^2)$. To address memory bottlenecks in LAVA for large datasets, Kessler et al. [32] propose SAVA, a batching strategy for Wasserstein computation, though other challenges remain unresolved.

**Dataset Valuation**   Another line of work focuses on valuing entire datasets rather than individual points, typically in settings with multiple data providers where fair compensation is desired [7, 3]. These methods use distance-based metrics such as mutual information [7], $\text{MMD}^2$ [62], MMD [74], and volume [75] to assess utility. However, they do not extend naturally to individual valuation due to their reliance on large datasets, inability to capture point-level interactions, and lack of influence estimation. Unlike these methods that compute distances between entire datasets, our approach quantifies individual datapoints' contributions to distributional distance through influence functions. This value is not equal to the distance between the individual point and the reference dataset.

# B  Proofs

## B.1   Proof of Proposition 1

**Proposition 1.** *The influence function for MMD as the distance metric is, up to additive and positive multiplicative constants, given by*

$$\text{IF}_{\text{MMD}}(x; P, Q) = \mathbb{E}_{x' \sim P}[k(x', x)] - \mathbb{E}_{x' \sim Q}[k(x', x)].$$

*Proof.* We first derive the influence function for $\text{MMD}^2$, then apply the chain rule to obtain the result for $\text{MMD}$.

$$
\begin{aligned}
\text{IF}_{\text{MMD}^2}(x; P, Q) &= -\left.\frac{d}{d\varepsilon}\text{MMD}^2(P, (1-\varepsilon)Q + \varepsilon\delta_x)\right|_{\varepsilon=0} \\
&= -\left.\frac{d}{d\varepsilon}\left\|\mu_P - ((1-\varepsilon)\mu_Q + \varepsilon\phi(x))\right\|_{\mathcal{H}}^2\right|_{\varepsilon=0} \\
&= -\left.\frac{d}{d\varepsilon}\left[\|\mu_P - \mu_Q\|_{\mathcal{H}}^2 + 2\varepsilon\langle\mu_P - \mu_Q, \mu_Q - \phi(x)\rangle + \varepsilon^2\|\mu_Q - \phi(x)\|^2\right]\right|_{\varepsilon=0} \\
&= -2\langle\mu_P - \mu_Q, \mu_Q - \phi(x)\rangle \\
&= 2\left(-\langle\mu_P, \mu_Q\rangle + \langle\mu_P, \phi(x)\rangle + \langle\mu_Q, \mu_Q\rangle - \langle\mu_Q, \phi(x)\rangle\right) \\
&= 2\left(\mathbb{E}_{x'\sim P}[k(x', x)] - \mathbb{E}_{x'\sim Q}[k(x', x)]\right) \\
&\quad - 2\left(\mathbb{E}_{x',x''\sim P,Q}[k(x', x'')] + \mathbb{E}_{x',x''\sim Q}[k(x', x'')]\right).
\end{aligned}
$$

Now, applying the chain rule:

$$
\text{IF}_{\text{MMD}}(x; P, Q) = \frac{\text{IF}_{\text{MMD}^2}(x; P, Q)}{2\text{MMD}(P, Q)}
$$

Ignoring terms independent of $x$, we obtain the simplified expression:

$$
\text{IF}_{\text{MMD}}(x; P, Q) = \mathbb{E}_{x'\sim P}[k(x', x)] - \mathbb{E}_{x'\sim Q}[k(x', x)] \tag{15}
$$

$\square$

### B.2 Proof of Proposition 2

**Proposition 2.** *Let $D^{train}$ and $D^{val}$ be finite samples from distributions $Q$ and $P$, respectively. If for all subsets $S \subseteq D^{train} \setminus \{x_i^{\text{train}}, x_j^{\text{train}}\}$,*

$$
\widehat{MMD}(D^{val}, S \cup \{x_i^{\text{train}}\}) - \widehat{MMD}(D^{val}, S) = \widehat{MMD}(D^{val}, S \cup \{x_j^{\text{train}}\}) - \widehat{MMD}(D^{val}, S)
$$

*then $\widehat{\text{IF}}(x_i^{\text{train}}) = \widehat{\text{IF}}(x_j^{\text{train}})$.*

*Proof.* Consider $S = D^{\text{train}} \setminus \{x_i^{\text{train}}, x_j^{\text{train}}\}$. We have,

$$
\widehat{\text{MMD}}(D^{\text{val}}, S \cup \{x_i^{\text{train}}\}) - \widehat{\text{MMD}}(D^{\text{val}}, S) = \widehat{\text{MMD}}(D^{\text{val}}, S \cup \{x_j^{\text{train}}\}) - \widehat{\text{MMD}}(D^{\text{val}}, S).
$$

Adding $\widehat{\text{MMD}}(D^{\text{val}}, S)$ to both sides gives

$$
\widehat{\text{MMD}}(D^{\text{val}}, S \cup \{x_i^{\text{train}}\}) = \widehat{\text{MMD}}(D^{\text{val}}, S \cup \{x_j^{\text{train}}\}).
$$

Squaring both sides,

$$
\widehat{\text{MMD}}^2(D^{\text{val}}, S \cup \{x_i^{\text{train}}\}) = \widehat{\text{MMD}}^2(D^{\text{val}}, S \cup \{x_j^{\text{train}}\}).
$$

Using the finite-sample estimator for $\widehat{\text{MMD}}$,

$$
\begin{aligned}
\widehat{\text{MMD}}^2(D^{\text{val}}, T) &= \frac{1}{n_{\text{val}}^2}\sum_{k,l} k(x_k^{\text{val}}, x_l^{\text{val}}) + \frac{1}{|T|^2}\sum_{k,l\in T} k(x_k^{\text{train}}, x_l^{\text{train}}) \\
&\quad - \frac{2}{n_{\text{val}}|T|}\sum_{k=1}^{n_{\text{val}}}\sum_{l\in T} k(x_k^{\text{val}}, x_l^{\text{train}})
\end{aligned}
$$

where $T = S \cup \{x_i^{\text{train}}\}$ or $T = S \cup \{x_j^{\text{train}}\}$. Therefore, substituting the finite-sample estimator into,

$$
\widehat{\text{MMD}}^2(D^{\text{val}}, S \cup \{x_i^{\text{train}}\}) = \widehat{\text{MMD}}^2(D^{\text{val}}, S \cup \{x_j^{\text{train}}\}).
$$

After canceling common terms and using $|T| = n_{\text{train}} - 1$,

$$\frac{2}{n_{\text{val}}} \sum_{m=1}^{n_{\text{val}}} k(x_k^{\text{val}}, x_i^{\text{train}}) - \frac{2}{n_{\text{train}} - 1} \sum_{l=1, l \neq i}^{n_{\text{train}}} k(x_l^{\text{train}}, x_i^{\text{train}})$$

$$= \frac{2}{n_{\text{val}}} \sum_{m=1}^{n_{\text{val}}} k(x_k^{\text{val}}, x_j^{\text{train}}) - \frac{2}{n_{\text{train}} - 1} \sum_{l=1, l \neq j}^{n_{\text{train}}} k(x_l^{\text{train}}, x_j^{\text{train}})$$

Therefore,

$$\widehat{\text{IF}}(x_i^{\text{train}}) = \widehat{\text{IF}}(x_j^{\text{train}})$$

$\square$

### B.3 Proof and Example of Proposition 3

**Intuition:** The core idea is that a threshold can be used to distinguish points where $P$ locally dominates $Q$ versus those where $Q$ dominates $P$, based on their relative densities in a neighborhood around each point. For accurate local density estimation, the kernel bandwidth $\sigma$ (e.g., in a Gaussian kernel) should be chosen sufficiently small. This result holds in the infinite sample regime where expectations are exact. In the finite sample case, this separation may not hold strictly, as small values of $\sigma$ can induce high-variance estimates.

**Proposition 3.** *Let $P$ and $Q$ be two probability distributions on $\mathcal{X} \subseteq \mathbb{R}^n$. For any $\epsilon > 0$ and $r > 0$, there exists a Gaussian isotropic kernel $k$ such that for $\text{IF}(x) = \mathbb{E}_{x' \sim P}\big[k(x, x')\big] - \mathbb{E}_{x' \sim Q}\big[k(x, x')\big]$, the following holds*

$$\begin{cases} \text{IF}(x) > 0, & \text{if } P(x') - Q(x') \geq \epsilon \, \forall \, x' \in B(x, r), \\ \text{IF}(x) < 0, & \text{if } P(x') - Q(x') \leq -\epsilon \, \forall \, x' \in B(x, r), \end{cases}$$

*where $B(x, r) = \{x' : \|x' - x\| < r\}$.*

*Proof.* Let

$$k(x, x') = \frac{1}{(2\pi\sigma^2)^{n/2}} \exp\Big(-\frac{\|x - x'\|^2}{2\sigma^2}\Big),$$

then

$$\text{IF}(x) = \int_{x' \in \mathbb{R}^n} (P(x') - Q(x')) \, k(x, x') \, dx' = \int_{x' \in B(x, r)} (P - Q) \, k(x, x') + \int_{x' \in B(x, r)^c} (P - Q) \, k(x, x').$$

Since $P(x') - Q(x') \geq \epsilon$ for all $x' \in B(x, r)$ and $|P - Q| \leq 1$ outside,

$$f(x) \geq \epsilon \int_{x' \in B(x, r)} k(x, x') - \int_{x' \in B(x, r)^c} k(x, x') = \epsilon A - (1 - A) = (\epsilon + 1)A - 1,$$

where

$$A = \int_{x' \in B(x, r)} k(x, x') \, dx'$$

Note that,

$$\int_{B(x, r)} k(x, x') \, dx' = \Pr_{X \sim \mathcal{N}(0, \sigma^2 I)} (\|X\| < r) = \Pr(\|X\|^2 < r^2) = 1 - \Pr(\|X\|^2 \geq r^2).$$

Let $W = \|X\|^2 = \sum_{i=1}^n X_i^2$, so $\mathbb{E}[W] = n\sigma^2$ and $\text{Var}(W) = 2n\sigma^4$. By Chebyshev's inequality, for $r^2 > n\sigma^2$,

$$\Pr(W \geq r^2) \leq \frac{\text{Var}(W)}{(r^2 - \mathbb{E}[W])^2} = \frac{2n\sigma^4}{(r^2 - n\sigma^2)^2},$$

hence

$$A \geq 1 - \frac{2n\sigma^4}{(r^2 - n\sigma^2)^2}.$$

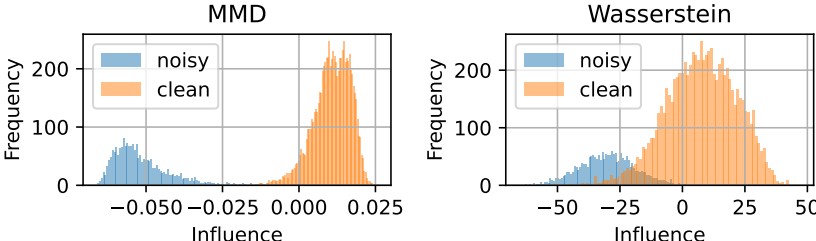

Figure 9: Influence distribution obtained from influence based on MMD (KAIROS) and Wasserstein (LAVA).

Choose $\frac{r}{\sqrt{n}} > \sigma^* > 0$ so that

$$1 - \frac{2n\sigma^{*4}}{(r^2 - n\sigma^{*2})^2} > \frac{1}{\epsilon + 1}.$$

Then $(\epsilon + 1)A - 1 > 0$, implying $\text{IF}(x) > 0$.

The same $\sigma^*$ gives

$$\text{IF}(x) \leq -\big((\epsilon + 1)A - 1\big) < 0$$

for the other case where $P(x') - Q(x') \leq -\epsilon$.

Thus, there exists a Gaussian kernel with bandwidth $\sigma^*$ that satisfies the desired separation. $\qquad\square$

**Example 1** (Density Separation). *We compare the influence distributions obtained from MMD and Wasserstein, computed on the CIFAR-10 sample data, to show the density separation property. As shown in Figure 9, MMD-based influence exhibits a near-zero optimal threshold that separates the noisy and clean points almost perfectly. On the other hand, the Wasserstein-based influence for noisy and clean data is entangled, and there is no threshold which cleanly separates the two.*

### B.4  Proof of Proposition 4

**Proposition 4.** *The influence function for* E-MCMD*, up to additive and positive multiplicative constants, given by*

$$\text{IF}_{\text{E-MCMD}}((x, y); P, Q) = \|\mu^P_{Y|X}(x) - \phi(y)\|_{\mathcal{H}_{\mathcal{Y}}} \tag{16}$$

*Proof.* Consider the perturbed distribution $Q_\varepsilon = (1 - \varepsilon)Q + \varepsilon\delta_{(x,y)}$, with corresponding marginal $(1 - \varepsilon)Q_X + \varepsilon\delta_x$. Since $Q_\varepsilon(Y = y \mid x) = 1$, the perturbed conditional embedding at point $x$ is given by

$$\mu^{Q_\varepsilon}_{Y|X}(x) = \phi(y),$$

Therefore,

$$\text{MCMD}_{P, Q_\varepsilon}(x') = \begin{cases} \text{MCMD}_{P,Q}(x') & \text{if } x' \neq x \\ \|\mu^P_{Y|X}(x) - \phi(y)\|_{\mathcal{H}_{\mathcal{Y}}} & \text{if } x' = x \end{cases} \tag{17}$$

E-MCMD$(P, Q_\varepsilon)$ can be written as:

$$\mathbb{E}_{X \sim (1-\varepsilon)Q_X + \varepsilon\delta_x}[\text{MCMD}_{P,(1-\varepsilon)Q + \varepsilon\delta_{(x,y)}}(X)]$$
$$= (1 - \varepsilon)\left(\mathbb{E}_{X \sim Q_X}[\text{MCMD}_{P,Q_\varepsilon}(X)]\right) + \varepsilon\left(\text{MCMD}_{P,Q_\varepsilon}(x)\right)$$
$$= (1 - \varepsilon)\left(\mathbb{E}_{X \sim Q_X}[\text{MCMD}_{P,Q}(X)]\right) + \varepsilon\|\mu^P_{Y|X}(x) - \phi(y)\|_{\mathcal{H}_{\mathcal{Y}}}$$

Therefore we have,

$$\text{IF}_{\text{E-MCMD}}((x,y); P, Q)$$

$$= -\lim_{\varepsilon \to 0^+} \frac{\mathbb{E}_{X \sim (1-\varepsilon)Q_X + \varepsilon\delta_x}[\text{MCMD}_{P,(1-\varepsilon)Q+\varepsilon\delta_x}(X)] - \mathbb{E}_{X \sim Q_X}[\text{MCMD}_{P,Q}(X)]}{\varepsilon}$$

$$= -\lim_{\varepsilon \to 0^+} \frac{(1-\varepsilon)\left(\mathbb{E}_{X \sim Q_X}[\text{MCMD}_{P,Q}(X)]\right) + \varepsilon\|\mu^P_{Y|X}(x) - \phi(y)\|_{\mathcal{H}_{\mathcal{Y}}} - \mathbb{E}_{X \sim Q_X}[\text{MCMD}_{P,Q}(X)]}{\varepsilon}$$

$$= -\lim_{\varepsilon \to 0^+} \frac{\varepsilon\|\mu^P_{Y|X}(x) - \phi(y)\|_{\mathcal{H}_{\mathcal{Y}}} - \varepsilon\mathbb{E}_{X \sim Q_X}[\text{MCMD}_{P,Q}(X)]}{\varepsilon}$$

$$= -\|\mu^P_{Y|X}(x) - \phi(y)\|_{\mathcal{H}_{\mathcal{Y}}} + \mathbb{E}_{X \sim Q_X}[\text{MCMD}_{P,Q}(X)]$$

Ignoring terms independent of $x$ and $y$,

$$\text{IF}_{\text{E-MCMD}}((x,y); P, Q) = -\|\mu^P_{Y|X}(x) - \phi(y)\|_{\mathcal{H}_{\mathcal{Y}}}$$

$\square$

## B.5 Proof of Theorem 1

We present the full statement of Theorem 1 below, followed by a detailed proof.

**Theorem 1** (Bounding transfer loss)**.** *Let $(\mathcal{X}, d_{\mathcal{X}})$ and $(\mathcal{Y}, d_{\mathcal{Y}})$ be compact metric spaces, and let $k_{\mathcal{X}}, k_{\mathcal{Y}}$ be universal kernels on $\mathcal{X}, \mathcal{Y}$ with RKHS $\mathcal{H}_{\mathcal{X}}, \mathcal{H}_{\mathcal{Y}}$. Equip $\mathcal{Z} = \mathcal{X} \times \mathcal{Y}$ with the tensor-product kernel whose RKHS is*

$$\mathcal{H} = \mathcal{H}_{\mathcal{X}} \widehat{\otimes} \mathcal{H}_{\mathcal{Y}}.$$

*Let $P, Q \in \mathcal{P}(\mathcal{Z})$ have marginals $P_X, Q_X \in \mathcal{P}(\mathcal{X})$ and conditionals $P(\cdot \mid x), Q(\cdot \mid x) \in \mathcal{P}(\mathcal{Y})$. Let $L : \mathcal{Z} \to \mathbb{R}$ be a continuous loss function. Since $\mathcal{H}$ is endowed with a universal kernel, $\mathcal{H}$ is dense in $C(\mathcal{Z})$. Let*

$$\|L\|_* = \inf_{L' \in \mathcal{H}} \left\{ \left\|\mathbb{E}_{y \sim P(\cdot|x)} L'(x,y)\right\|_{\mathcal{H}_{\mathcal{X}}} + \sup_{x \in \mathcal{X}} \left\|\sup_{y \in \mathcal{Y}} L'(x,y)\right\|_{\mathcal{H}_{\mathcal{Y}}} + \|L - L'\|_\infty \right\}.$$

*Then*

$$\mathbb{E}_{(x,y) \sim Q}[L(x,y)] \leq \mathbb{E}_{(x,y) \sim P}[L(x,y)] + \|L\|_* \left( \text{MMD}_{\mathcal{X}}(P_X, Q_X) + \mathbb{E}_{x \sim Q_X}\left[\text{MCMD}_{P,Q}(x)\right] \right).$$

*Proof.* Fix $\varepsilon > 0$. Since $\mathcal{H}$ is dense in $C(\mathcal{Z})$, choose $L' \in \mathcal{H}$ with $\|L - L'\|_\infty \leq \varepsilon$. Then

$$\mathbb{E}_Q[L] = \mathbb{E}_Q[L'] + \mathbb{E}_Q[L - L'], \qquad \mathbb{E}_P[L] = \mathbb{E}_P[L'] + \mathbb{E}_P[L - L'],$$

so

$$\mathbb{E}_Q[L] - \mathbb{E}_P[L] = \left(\mathbb{E}_Q[L'] - \mathbb{E}_P[L']\right) + \left(\mathbb{E}_Q[L - L'] - \mathbb{E}_P[L - L']\right).$$

Since $P$ and $Q$ are probability measures,

$$\left|\mathbb{E}_Q[L - L'] - \mathbb{E}_P[L - L']\right| \leq \|L - L'\|_\infty \leq \varepsilon.$$

By the law of total expectation,

$$\mathbb{E}_Q[L'] = \mathbb{E}_{x \sim Q_X}\left[\mathbb{E}_{y \sim Q(\cdot|x)} L'(x,y)\right], \quad \mathbb{E}_P[L'] = \mathbb{E}_{x \sim P_X}\left[\mathbb{E}_{y \sim P(\cdot|x)} L'(x,y)\right].$$

Add and subtract $\mathbb{E}_{x \sim Q_X} \mathbb{E}_{y \sim P(\cdot|x)} L'$ to obtain

$$\mathbb{E}_Q[L'] - \mathbb{E}_P[L'] = A + B,$$

where

$$A = \mathbb{E}_{x \sim Q_X}\left[\mathbb{E}_{y \sim Q(\cdot|x)} L'(x,y) - \mathbb{E}_{y \sim P(\cdot|x)} L'(x,y)\right],$$

$$B = \mathbb{E}_{x \sim Q_X}\left[\mathbb{E}_{y \sim P(\cdot|x)} L'(x,y)\right] - \mathbb{E}_{x \sim P_X}\left[\mathbb{E}_{y \sim P(\cdot|x)} L'(x,y)\right].$$

Define $f_{L'}(x) = \mathbb{E}_{y \sim P(\cdot|x)} L'(x,y) \in \mathcal{H}_{\mathcal{X}}$. Then

$$B = \mathbb{E}_{x \sim Q_X} f_{L'}(x) - \mathbb{E}_{x \sim P_X} f_{L'}(x),$$

so
$$|B| \leq \|f_{L'}\|_{\mathcal{H}_{\mathcal{X}}} \mathrm{MMD}_{\mathcal{X}}(P_X, Q_X) = \|L'\|_{\mathcal{H}}^{(\mathcal{X})} \mathrm{MMD}_{\mathcal{X}}(P_X, Q_X).$$

For each $x$, let $L'_y(x, \cdot) \in \mathcal{H}_{\mathcal{Y}}$. Then

$$\left| \mathbb{E}_{y \sim Q(\cdot|x)} L'(x, y) - \mathbb{E}_{y \sim P(\cdot|x)} L'(x, y) \right| \leq \|L'_y(x, \cdot)\|_{\mathcal{H}_{\mathcal{Y}}} \mathrm{MCMD}_{P,Q}(x),$$

hence

$$|A| \leq \mathbb{E}_{x \sim Q_X} \left[ \|L'\|_{\mathcal{H}}^{(\mathcal{Y})} \mathrm{MCMD}_{P,Q}(x) \right] = \|L'\|_{\mathcal{H}}^{(\mathcal{Y})} \mathbb{E}_{x \sim Q_X}[\mathrm{MCMD}_{P,Q}(x)].$$

Combining the bounds for $A$, $B$, and the approximation term,

$$\mathbb{E}_Q[L] \leq \mathbb{E}_P[L] + \|L'\|_{\mathcal{H}}^{(\mathcal{X})} \mathrm{MMD}_{\mathcal{X}}(P_X, Q_X) + \|L'\|_{\mathcal{H}}^{(\mathcal{Y})} \mathbb{E}_{x \sim Q_X}[\mathrm{MCMD}_{P,Q}(x)] + \varepsilon.$$

Taking the infimum over $L' \in \mathcal{H}$ yields the desired result. $\qquad\square$

## C  Influence Computation and Update for Streaming data

Algorithm 1 provides the detailed algorithm for computing the influence in the online setting.

---
**Algorithm 1** Online update for influence

---
**Input:** Current size $n_t$, variables $\mathcal{A}^{(t)}, \mathcal{B}^{(t)}, \mathcal{R}^{(t)} \in \mathbb{R}^{n_t}$, validation size $n_{\mathrm{val}}$, batch size $m$, incoming batch data ($X^{\mathrm{new}} \in \mathbb{R}^{m \times d}$, $Y^{\mathrm{new}} \in \mathbb{R}^{m \times c}$), balancing factor $\lambda$.
**Output:** Updated $\mathcal{A}^{(t+1)}, \mathcal{B}^{(t+1)}, \mathcal{R}^{(t+1)}, \mathcal{V}^{(t+1)} \in \mathbb{R}^{n_{t+1}}$.

1: $n_{t+1} \leftarrow n_t + m$
2: compute $K^{\mathrm{old,new}} \in \mathbb{R}^{n_t \times m}$, $K^{\mathrm{new,old}} \in \mathbb{R}^{m \times n_t}$, $K^{\mathrm{new,new}} \in \mathbb{R}^{m \times m}$   // kernel sub-matrices
3: **for** $i = 1, \ldots, n_t$ **do**   // update kernel means for old points
4:     $\mathcal{A}_i^{(t+1)} \leftarrow \frac{1}{n_{t+1}-1}\left((n_t - 1)\mathcal{A}_i^{(t)} + \sum_{j=1}^m K_{i,j}^{\mathrm{old,new}}\right)$
5:     $\mathcal{B}_i^{(t+1)} \leftarrow \mathcal{B}_i^{(t)}$
6:     $\mathcal{R}_i^{(t+1)} \leftarrow \mathcal{R}_i^{(t)}$
7: **end for**
8: $\mathcal{R}_i^{\mathrm{new}} = \left\|y_{n_t+i}^{\mathrm{train}} - \hat{y}_{n_t+i}^{\mathrm{train}}\right\|$   $(i = 1, \ldots, m)$   // residuals for new points
9: **for** $i = 1, \ldots, m$ **do**   // update kernel means for new points
10:     $\mathcal{A}_{n_t+i}^{(t+1)} \leftarrow \frac{1}{n_{t+1}-1}\left(\sum_{j=1}^{n_t} K_{i,j}^{\mathrm{new,old}} + \sum_{j'=1}^m K_{i,j'}^{\mathrm{new,new}} - 1\right)$
11:     $\mathcal{B}_{n_t+i}^{(t+1)} \leftarrow \frac{1}{n_{\mathrm{val}}} \sum_{j=1}^{n_{\mathrm{val}}} k\left(x_j^{\mathrm{val}}, x_{n_t+i}^{\mathrm{train}}\right)$
12:     $\mathcal{R}_{n_t+i}^{(t+1)} \leftarrow \mathcal{R}_i^{\mathrm{new}}$
13: **end for**
14: **for** $i = 1, \ldots, n_{t+1}$ **do**
15:     $\mathcal{V}_i^{(t+1)} \leftarrow \lambda\left(\mathcal{B}_i^{(t+1)} - \mathcal{A}_i^{(t+1)}\right) + (1-\lambda)\mathcal{R}_i^{(t+1)}$   // net influence
16: **end for**

---

## D  Additional Experiments and Detailed Settings

**Detailed description of baselines and choice of hyper-parameters.**   We compare with the following baselines: LAVA [30] is a model-agnostic method that uses the Wasserstein-1 dual potential as an influence proxy; DATAOOB [36] trains a bootstrap ensemble and assigns each point its average out-of-bag loss to capture its contribution; DVRL [76] learns per-sample importance weights via reinforcement learning and uses those weights directly as data scores; and KNNSHAPLEY [27] computes exact Shapley values on a nearest-neighbor proxy model, exploiting its closed-form solution for efficiency. To avoid the prohibitive runtime of DATAOOB without sacrificing accuracy, we decrease its number of bootstraps from 1000 to 100. All baselines run with their default settings in OpenDataVal [29].

Following [19, 6, 23], we adopt the "median heuristic" to set the kernel bandwidth to the median of pair-wise distances. Given the large number of samples, we estimate the median on 10000 sampled pairs. In practice, as shown in experiments, sampled bandwidths work well across various datasets and scenarios. The balancing factor is determined by aligning the scale of the two components of net influence (Equation (12)), when both feature and label noise exist.

**Comparison with Sample Reweighting Methods**    We compare KAIROS with MAPLE [78], a model-based sample reweighting method. Unlike KAIROS, MAPLE does not compute data values but learns reweighting functions for specific training objectives. While applicable to corrupted label detection, KAIROS offers broader applicability across tasks such as detecting harmful fine-tuning data and data poisoning attacks. MAPLE is not model-agnostic and inherits the limitations of model-based approaches discussed in Section A.

We evaluate both methods on CIFAR-10 feature noise and label noise detection. For MAPLE, we use target labels as group labels since no explicit group labels are available. We report the AUC of the fraction of covered corrupted data versus the fraction of inspected data.

Table 3: AUC for CIFAR-10 feature noise and label noise detection tasks.

| Method | AUC Feature Noise | AUC Label Noise |
|---|---|---|
| Data OOB | 0.727 | 0.784 |
| KNN Shapley | 0.723 | 0.751 |
| LAVA | 0.837 | 0.529 |
| KAIROS (Gaussian) | 0.857 | 0.791 |
| MAPLE | 0.347 | 0.828 |
| Maximum possible | 0.900 | 0.900 |

MAPLE performs well on label noise detection but poorly on feature noise detection. While reweighting methods can be effective for specific corruption types like label noise, KAIROS provides consistent performance across diverse tasks including feature noise, label noise, adversarial attacks, and harmful fine-tuning detection.

**Selection Bias Detection**    We evaluate KAIROS on selection bias where subgroups are under-represented in training data. We use the ACS Income dataset from WhyShift [**?** ], which predicts income from demographic attributes. This dataset exhibits geographic bias where some states like Puerto Rico (PR) are under-represented while others like California (CA) are over-represented. Liu et al. [40] show that models trained predominantly on CA data fail to generalize to PR.

We simulate selection bias by creating a training set with 80% CA and 20% PR samples (1000 total) and a balanced validation set with 50% CA and 50% PR samples (300 total). We evaluate how well KAIROS and baselines identify points from the under-represented group (PR). Data valuation methods should assign high influence to under-represented samples. We report the AUC of the fraction of under-represented samples detected versus the top $k$ fraction of data chosen.

Table 4: Selection bias detection performance on ACS Income dataset.

| Method | AUC |
|---|---|
| Data OOB | 0.326 |
| KNN Shapley | 0.494 |
| LAVA | 0.546 |
| KAIROS | 0.855 |
| Maximum possible (theoretical) | 0.900 |

The results show that KAIROS considerably outperforms both model-agnostic and model-based baselines in identifying under-represented samples. For data redundancy, KAIROS rankings remain stable when datasets contain duplicates. We verified this by duplicating the training dataset up to 10 times and confirming that relative rankings are preserved.

**Comparison with additional Model-based Methods**    We compare KAIROS with TRAK [46] and LOGRA [8], recent model-based attribution methods for large models. TRAK trains multiple models while LOGRA optimizes influence functions for efficiency. Both require model training, while KAIROS is model-agnostic and training-free.

We evaluate on CIFAR-10 feature noise and label noise detection using the experimental setup from Section 4. We report the AUC of the fraction of covered corrupted data versus the fraction of inspected data.

Table 5: AUC comparison on CIFAR-10 corruption detection tasks.

| Method | Feature Noise | Label Noise |
|---|---|---|
| Data OOB | 0.727 | 0.784 |
| KNN Shapley | 0.723 | 0.751 |
| LAVA | 0.837 | 0.529 |
| KAIROS | 0.857 | 0.791 |
| TRAK | 0.638 | 0.743 |
| LOGRA | 0.565 | 0.663 |
| Maximum possible (theoretical) | 0.900 | 0.900 |

KAIROS achieves the highest AUC on both tasks, outperforming TRAK and LOGRA particula

**Effect of kernel choice** In order to study the effect of kernel choice, we conduct additional experiments using a polynomial kernel of degree 2 on the feature-noise CIFAR-10 task. We report the AUC of the fraction of covered corrupted data vs the fraction of inspected data.

Table 6: AUC for detecting feature noise in CIFAR-10

| Method | AUC |
|---|---|
| Data OOB | 0.727 |
| KNN Shapley | 0.723 |
| LAVA | 0.837 |
| KAIROS (Gaussian) | **0.857** |
| KAIROS (Polynomial) | 0.856 |
| Maximum possible (theoretical) | 0.900 |

The results show that KAIROS with polynomial kernels achieves nearly identical performance to Gaussian kernels (0.856 vs 0.857) and outperforms all baselines.

**Estimating $P(y|x)$ from Validation Data** The validation set is typically small (300 samples in our experiments), yielding noisy $P(y|x)$ estimates. However, this provides sufficient signal to identify corrupted training points. After removing these points, we train on the cleaned, larger training set to obtain better $P(y|x)$ estimates and higher accuracy.

We demonstrate this on CIFAR-10 with feature noise. A model trained only on validation data (300 samples) achieves 72% test accuracy. Training on the full training set after removing the bottom 20% identified by KAIROS achieves 88.5% test accuracy. Even selecting just the top 300 training samples (matching validation size) gives 87.3% accuracy, demonstrating that KAIROS improves sample quality beyond what the small validation set provides. This shows that while the validation classifier has limited accuracy, it successfully identifies low-quality training data, enabling substantial improvements when training on the cleaned set.

**Robustness to noise in the validation set** KAIROS, similar to LAVA, assumes access to a small sample from the (clean) reference distribution. Further, model-based methods value data based on performance on the validation set which is generally assumed to be representative of the test distribution.

However, in practice, these small samples of the reference set may have some noise. We test the robustness of KAIROS and other baselines to a noisy reference set. We conduct additional experiments by adding noise to the validation set for the feature noise CIFAR-10 task. We consider two settings where we randomly corrupt 3% and 7% of the validation samples respectively.

**Dataset Licensing Information.** The license terms for each dataset used in this work are as follows:

Table 7: Robustness to Noisy Reference Set - AUC Performance

| Method | No Validation Noise | 3% Validation Noise | 7% Validation Noise |
|---|---|---|---|
| Data OOB | 0.727 | 0.711 | 0.711 |
| KNN Shapley | 0.723 | 0.705 | 0.705 |
| LAVA | 0.837 | 0.635 | 0.601 |
| KAIROS | **0.857** | **0.857** | **0.856** |

- CIFAR-10 [34]: released under the MIT License. `https://www.cs.toronto.edu/~kriz/cifar.html`

- STL-10 [10]: no explicit license is provided. `http://ai.stanford.edu/~acoates/stl10/`

- IMDB [43]: subject to IMDb's non-commercial terms of use. `https://datasets.imdbws.com/`

- AG News [77]: provided for research and non-commercial use. `http://www.di.unipi.it/~gulli/AG_corpus_of_news_articles.html`

