# OpenReview forum: "KAIROS: Scalable Model-Agnostic Data Valuation"
_NeurIPS.cc/2025/Workshop/Reliable_ML — NeurIPS 2025 - Reliable ML Workshop_

### Official Review · Reviewer_rrtS · 2025-09-19
**Model-Agnostic Data Valuation Which Can Help to Detect Distribution Shift**

**Rating:** 8
**Confidence:** 3

**Review:**

### Summary:
The author's derive a new method to evaluate data in a model-agnostic way  using a distributional influence score, similar to Wasserstein-based models for data valuation. Their score captures the maximum mean discrepancy (MMD) between the training data and a clean reference, and they derive a closed-form influence function which closely matches Leave-one-out rankings. Their method provides a natural threshold to separate low- and high-quality data and as they do not rely on model based methods or iterative optimisation, the method is easy to implement also for large scale datasets. Empirically, it compares favorably to Wasserstein-based and other existing approaches.
### Strength:
A theoretically grounded framework for an important practical problem, with clear empirical evidence. The approach is fast (including in online settings) and easy to deploy, and the experiments systematically probe data corruption/noise and the effect of pruning “bad” (and “good”) points on downstream performance. Their method also allows for detection of label-specific corruption, making e.g. clean label attacks detectable.
### Weaknesses:
The experiments demonstrate feasibility with fixed number of clean references, and Appendix Figure 6 and discussion explores the effect of validation-set size. However, the sample-size dependence remains implicit throughout the paper. It would help to add theoretical guarantees or general guidance on how validation-set size relates to accuracy (and possibly to data/label complexity), so practitioners can budget the number of clean examples needed.
### Suggestions:
Adress above weakness.

---

### Official Review · Reviewer_ZoUQ · 2025-09-20
**KAIROS is an efficient influence-based data valuation method**

**Rating:** 5
**Confidence:** 4

**Review:**

## Summary
The paper introduces KAIROS, a scalable, model-agnostic framework for data valuation. KAIROS measures each training point’s contribution to the Maximum Mean Discrepancy (MMD) between the training distribution and a clean reference set. The authors derive a closed-form influence function and design efficient batch and streaming algorithms.

## Strengths
1. Theoretical discussions and proofs.
2. Efficient and scalable design. Consider both offline and online streaming cases.
3. Comprehensive empirical validaitons.

## Weaknesses
- Rely on a clean validation/reference distribution. This is not realistic in large-scale or deployment settings. In comparison, memorization methods don't require a separate clean distribution.
- Missed comparison with memorization/influence methods. Although the motivation is different, memorization or per-sample influence methods are closely related concepts [1, 2]. These can serve directly as baselines for data valuation.


## Suggestions
1.  Line 43: missing citation


[1] Grosse, Roger, et al. "Studying large language model generalization with influence functions." arXiv preprint arXiv:2308.03296 (2023).
[2] Ravikumar, Deepak, et al. "Towards memorization estimation: Fast, formal and free." Forty-second International Conference on Machine Learning. 2025.